# Probing the Skin–Brain Axis: New Vistas Using Mouse Models

**DOI:** 10.3390/ijms23137484

**Published:** 2022-07-05

**Authors:** Aliće Weiglein, Evelyn Gaffal, Anne Albrecht

**Affiliations:** 1Institute of Anatomy, Otto-von-Guericke-University, Leipziger Str. 44, 39120 Magdeburg, Germany; anne.albrecht@med.ovgu.de; 2Department of Dermatology, Otto-von-Guericke-University, Leipziger Str. 44, 39120 Magdeburg, Germany; evelyn.gaffal@med.ovgu.de; 3Center for Behavioral Brain Sciences (CBBS), Universitätsplatz 2, 39106 Magdeburg, Germany

**Keywords:** atopic dermatitis, psoriasis, skin inflammation, skin–brain axis, HPA axis, cytokines

## Abstract

Inflammatory diseases of the skin, including atopic dermatitis and psoriasis, have gained increasing attention with rising incidences in developed countries over the past decades. While bodily properties, such as immunological responses of the skin, have been described in some detail, interactions with the brain via different routes are less well studied. The suggested routes of the skin–brain axis comprise the immune system, HPA axis, and the peripheral and central nervous system, including microglia responses and structural changes. They provide starting points to investigate the molecular mechanisms of neuropsychiatric comorbidities in AD and psoriasis. To this end, mouse models exist for AD and psoriasis that could be tested for relevant behavioral entities. In this review, we provide an overview of the current mouse models and assays. By combining an extensive behavioral characterization and state-of-the-art genetic interventions with the investigation of underlying molecular pathways, insights into the mechanisms of the skin–brain axis in inflammatory cutaneous diseases are examined, which will spark further research in humans and drive the development of novel therapeutic strategies.

## 1. Introduction–Skin Disease and Mental Health

Atopic dermatitis (AD) and psoriasis are T-cell-mediated skin diseases driven by environmental and genetic factors and are characterized by different clinical and pathophysiological aspects. AD is a chronic recurrent inflammatory skin disease affecting about 10–30% of children and 3–10% of adults. It is associated with the genetic predisposition to enhanced Th2-dependent inflammatory immune response [1]. IL-4 and IL-13, for example, are cytokines in the blood that regulate signature aspects of type 2 inflammatory responses, for example, by acting on T cells, B cells, and macrophages. In T cells, IL-4 was found to be capable of inducing the differentiation of naïve CD4 T cells into Th2 cells and, in B cells, it modulates the immunoglobulin class switch to IgG1 and IgE. Both IL-4 and IL-13 induce macrophage activation and trigger signaling cascades, such as the STAT6 pathway and IRS molecule pathways, including P12K, Akt, PKBE, and mTOR [2]. These pathways lead to an impaired epidermal barrier function, including the occurrence of relapsing eczema, chronic dry skin, severe itching, and an increased tendency to develop cutaneous infections [1]. In about 30% of patients, AD precedes the development of allergic asthma and allergies, a phenomenon known as ‘atopic march’ [3,4]. Over the past decades, the number of affected patients has drastically increased in developed countries [3,5]. Psoriasis is rare in children, but it has a prevalence of 1–9% in adults with a peak in early adulthood [6]. In genome-wide association studies, over 60 disease susceptibility regions have been identified and revealed the central pathogenic involvement of type 17 T-helper (Th17)-dependent cell activation [7]. In contrast to AD, skin lesions are chronic scaling plaques with a clear demarcation. In addition to the characteristic disfiguring skin lesions, patients often suffer from systemic manifestations of chronic inflammation, e.g., in the cardiovascular system, intestine, joints, and bones. Therefore, they have a greater risk of developing concomitant diseases, such as high blood pressure, chronic inflammatory bowel disease, obesity, hyperglycemia, or psoriatic arthritis [8].

Common in atopic dermatitis and psoriasis is the association with psychosomatic illnesses, including anxiety, depression, or addictive behavior [9]. Atopic dermatitis often is related to increased stress, attention deficit hyperactivity disorder (ADHD), anxiety, depression, and suicidal ideation. In psoriasis, the proportion of alcoholics or depressive patients is significantly higher in comparison to other skin diseases [10,11,12]. One explanation is that distress related to active skin disease can lead to enhanced and continuous psychological stress with an increasing risk of psychiatric comorbidities. On the other hand, depressive episodes as well as acute or chronic stress in patients with unaffected skin can predate a flare-up of the skin disease. All this can lead into a vicious circle that impacts patient behavior and mental health [13,14,15].

Numerous studies have highlighted the bidirectional connection between systemic inflammation and psychiatric disease in general and identified intensive acute and chronic psychobiological stress as a risk factor of exacerbation in both domains [10]. Stress activates the hypothalamic–pituitary–adrenal (HPA) axis, which can control the immune system via neuroendocrine factors and the autonomous nervous system [16]. Many inflammatory mediators involved in the pathogenesis of AD and psoriasis are associated with major depression or anxiety disorders as well and affect brain function [17,18]. Recent findings further highlight the gut and skin microbiome as additional modulatory factors mediating skin and brain interactions [19].

In this review, we provide a brief overview of systems and putative molecular factors connecting the skin and brain in chronic skin inflammation and discuss current mouse models of cutaneous diseases. We then introduce behavioral assays to study neuropsychiatric comorbidities in these experimental models and suggest interventions for probing the molecular mechanisms of skin–brain axis interactions.

## 2. Modeling the Skin–Brain Axis

The relationship between psychological stress and inflammatory skin diseases seems to be bi-directional with stress exacerbating skin inflammation and skin inflammation leading to neuropsychiatric comorbidities. Potentially, this can result in a vicious cycle, in which acute stress negatively affects skin inflammation, which in turn raises anxiety [20]. While the involved systems have been reviewed extensively elsewhere, we provide in this paper a brief overview of the potential players in the skin–brain axis.

For an overview of the systems involved in an immune response, see Figure 1, and for central and peripheral responses to chronic stress, see Figure 2.

### 2.1. Effects of the HPA Axis on Skin Inflammation

During acute or chronic physical and psychological challenges, the central HPA axis is activated. In this case, the paraventricular nucleus of the hypothalamus releases the neuropeptide corticotropin-releasing hormone (CRH), which then causes the secretion of pro-opiomelanocortin (POMC)-derived peptides from the anterior pituitary gland. One of these peptides, the adrenocorticotropic hormone (ACTH), induces the release of glucocorticoid hormones (GCs) from the adrenal cortex ([21]; Figure 1). Numerous tissues express receptors for GCs (cortisol in humans and corticosterone in rodents), including neurons and astrocytes in the brain and keratinocytes in the skin and immune cells. GCs modulate skin inflammation together with other neuroendocrine mediators, such as histamine, via the H4 receptor and promote inflammation at physiological levels [2,22]. Importantly, under chronic stress, the negative feedback regulation of the HPA axis by GCs is disturbed. A similar dysregulation is also observed in AD patients and is linked to their altered immune response dominated type 2 T-helper (Th2) cells [2]. Moreover, chronic stress is correlated with a dysfunctional permeability barrier of the skin, in particular of the stratum corneum, which increases GC levels further due to a disturbed ‘peripheral HPA axis’ [20,23]. 

The peripheral HPA axis means that structures outside of the CNS are capable of producing the same molecular components as the central HPA axis, namely CRH, ACTH, and cortisol, as well as neurotransmitters, neuropeptides, and neurotrophins. In the epidermis, keratinocytes take over this role and produce CRH and GCs (Figure 1). CRH can, furthermore, be synthesized by immune cells, mast cells, and local nerve endings [24] and acts through the CRHR1 receptor on various cell types, including keratinocytes, mast cells, and melanocytes (see also Figure 2) [24,25]. In addition, CRH acts through CRH-R2 on blood vessels and modulates angiogenesis as well as vascular permeability [2], and it contributes to skin integrity by regulating sebaceous glands.

The second player, the autonomic system, acts via the sympathetic arm of the peripheral nervous system (PNS) on the skin. It is triggered by the locus coeruleus and the norepinephrine system of the central nervous system (CNS), resulting in the secretion of the catecholamines noradrenaline and adrenaline from nerve fiber terminals targeted, especially, in the dermis and subcutaneous fat layers [21]. Under acute stress, adrenaline and noradrenaline are released also systemically by the medullary part of the adrenal gland, leading to decreased skin blood flow as well as altered cytokine production and lymphocyte trafficking [26]. As the classical counterplayer of the sympathetic system, the parasympathetic system acts by sending cholinergic fibers to the skin that originate from the vagal nucleus of the brain stem. This nucleus bidirectionally interacts with the hypothalamus and the HPA axis and may affect skin integrity [26,27]. In addition to autonomous fibers, the skin, as our largest sensory organ, contains numerous receptors and free nerve endings, building afferents to the spinal cord. In this case, numerous neuropeptides have been found that modulate neurotransmission. Interestingly, during skin inflammation, abnormal patterns of cutaneous innervation and changes in the expression level of neuropeptides have been described [24]. One of these peptides, substance P (SP), activates keratinocytes and mast cells, which secrete histamine, cytokines, and nerve growth factors, all biologically relevant molecules for inflammatory processes [26]. 

### 2.2. Effects of Peripheral Inflammation on Brain Plasticity and Stress-Related Behavioral Domains

Studies in patient cohorts have demonstrated an association of the inflammatory mediators involved in the pathogenesis of AD and psoriasis, such as tumor necrosis factor (TNF)-α, interleukin (IL)-1, and IL-6, with depression or anxiety disorders [17,18]. Studies in children with atopic dermatitis revealed that early life overexposure to the Th2-cytokine IL-4 affects the developing brain and increases the risk to develop attention deficit hyperactivity disorder [28,29]. 

Under normal conditions, cytokines support plasticity, such as long-term potentiation and depression (LTP and LTD, respectively), as well as memory formation and behavioral domains (see Table 1 for important factors). Cytokines can reach the brain via the blood, being released from local immune cells in the skin tissue (Figure 2; [24]). The exact cellular mechanisms of cytokine action in the brain are diverse and currently best described for IL-1β, IL-6, and TNF-α as the most intensively studied examples. They induce changes in monoamine levels, such es norepinephrine, dopamine, and serotonin, as well as in the cholinergic system or opioid system. Changes in neuromodulator levels may then alter glutamate metabolism and the expression of the neurotransmitter receptors, such as NMDAR, AMPAR, and GABA. The induction of gene expression via cytokines is further achieved via the induction of growth factors, such as NGF and BDNF, which in turn control plasticity-relevant intracellular signaling cascades, such as the MAPK/ERK pathways, and activate transcription factors, such as cFos and CREB. Cytokines can also induce the transcription factor ‘kappa-light-chain-enhancer’ in activated B-cells that may, through a feedback loop, trigger cytokine production [30]. 

During skin inflammation, peripheral cytokine levels increase, with detrimental effects on neuronal health and plasticity. In this case, IL-1β, IL-6, and TNF-α can act synergistically and potentiate their neuronal impact, but also their respective expression levels [30]. Of note, during peripheral expression, but also in neurogenerative diseases and under psychological stress exposure, cytokines are also produced by a special population of macrophages located in the brain parenchyma, the microglia [30]. This effect is further enhanced by a crosstalk to the HPA axis, which then boosts peripheral and brain cytokine expression further, thereby additionally affecting neurotransmission, plasticity, and memory function (see Figure 2). 

Potentially, central nervous and peripheral cytokine interactions further complicating the picture of cytokine effects in the CNS. For example, changes in functional brain connectivity were found to be dependent on elevated IL-6 levels. Increased IL-6 in brain tissue, thereby, is linked to decreased functional connectivity between the ventral medial prefrontal cortex (mPFC) and the striatum, a pathway important for controlling addictive behavior. Increased IL-6 concentrations in the periphery, i.e., in serum, raised the functional connectivity between the dorsomedial prefrontal cortex (PFC) and the amygdala, an important circuit for controlling anxiety and emotional memory [30]. While the exact mechanisms of such a distinct circuit regulation remain obscure, region-specific shifts in brain activity are also observed under chronic skin inflammation. For example, the activity of the amygdala measured by positron emission tomography (PET) was elevated in psoriasis compared to healthy subjects and correlated with patient-reported depression and the severity of skin disease [31]. 

In summary, a shift in cytokine profiles leads to the over-activity of inflammatory molecules and malfunctions of the skin barrier, which, in turn, causes further changes in favor of sustained inflammatory activity. To date, a comprehensive model integrating the interactions of the various systems and the players involved is missing.

**Table 1 ijms-23-07484-t001:** Molecular players of the skin–brain axis. Involved cell types and their molecular substrates as well as the mechanism of action of molecular substrates, such as cytokines, along the skin–brain axis.

Active Substance	Site of Action	Function	Reference
IL-4	Astrocytes, neurons	↓ POMC expression; induces differentiation of CD4- into Th2 cells; macrophage activation	[2,32]
IL-13	Blood	Macrophage activation	[32]
TNF-α	Astrocytes, neurons	Astrocytic Ca2+ levels ↓ Glutamatergic exocytotic vesicles at the synapse p38–MAPK, ERK, JNK * pathways ↓ Spine size ↓ Glutamate receptors of the AMPA subtype	[2,30]
TSLP *****	Blood, skin	↑ Lymphoid cell response; CD4- T-cell polarization into Th2 cells	[33]
IL-1β	Neurons	Production of NGF **; BDNF *** release activation of the tropomyosin receptor kinase B (TrkB)–ERK pathway High level: ↓ LTP, spatial and working memory	[30,34,35]
IL-6	Neurons blood	ERK1/2 pathway High levels: ↓ LTD, ↓ memory, ↓ functional brain connectivity mPFC–striatum ↓ Functional brain connectivity mPFC–amygdala	[30]
IL-18	Blood, skin	↑ Th2 cytokines	[33]
IL-33	Blood/blood vessels	↑ Vascular endothelial growth factor (VEGF) release	[24]
NT ****	Blood/skin	↑ Histamine release from mast cells	[24]
CRH	Skin, blood, neurons,	↑ Activates mast cells ↑ HPA axis: POMC/ACTH/glucocorticoid levels	[24,25,36]
SP	Blood/skin	↑ Activates mast cells, ↑ Histamine, cytokines, NGF	[24]
VEGF	Blood/skin	Maturation of dendritic cells	[37]

* JNK: c-Jun N-terminal kinase; ** NGF: nerve growth factor; *** BDNF: brain-derived neurotrophic factor; **** NT: neurotensin; ***** TSLP: thymic stromal lymphopoietin.

### 2.3. Influences of a Neglected Regulatory System: The Microbiome

While the exact molecular mechanisms of an interaction between chronic inflammation and neuropsychiatric disorders are not completely clear, the microbiome as an important factor has attracted attention in the last decade. The microbiome describes the community of microorganisms in our bodies, comprising bacteria, fungi, virus, and single-cell organisms found on our skin and in high density, especially in the intestinal tract. There, they are best studied as key regulators of the gut–brain axis and have been firmly linked to several psychiatric diseases, for example, stress-related disorders, autism, anxiety, depression, and schizophrenia [38,39,40]. In addition, a disturbed microbiota–gut–brain (MGB) axis has been described for neurodegenerative diseases, such as Alzheimer’s and Parkinson’s diseases [41], and other conditions, such as obesity or irritable bowel syndrome [39]. Bacteria in the gut can directly alter neurotransmission in the vagus nerve and the enteric nervous system via microbial metabolites, such as amino acids, peptidoglycans, and serotonin metabolites [39]. Studies using germ-free animal rodents have described how several behavioral parameters, such as anxiety and depression, and how plasticity in various brain regions is affected by the MGB axis (see also [39] for a comprehensive review). Importantly, the composition of the gut microbiome can change under stress and an altered microbiome shifts HPA axis function, which, in turn, alters hormone levels and brain function. Moreover, the MGB axis possesses its own immunological system, with cytokine and chemokine release from enterocytes and a more specific immune response via lymphocytes of the gut-associated lymphoid tissue [39]. The MGB can affect systemic inflammation by modulating chronic inflammation directly via cytokines and indirectly via the HPA axis [38,42]. It is perfectly suited to mediate inflammatory effects towards the skin and the brain and, therefore, an important player for mediating the effects of cutaneous diseases on neuropsychiatric symptom complexes [19]. 

## 3. Tools for Probing the Skin–Brain Axis

To investigate the underlying pathobiological mechanisms and novel therapeutic approaches, animal models have been developed for AD and psoriasis. In order to expedite our knowledge about the interaction of inflammation and stress, these models are valuable tools. In parallel, many behavioral assays exist that allow us to assess endophenotypes relevant for neuropsychiatric disorders. However, such assays have barely been used in AD and psoriasis models. Therefore, we introduce both chronic skin inflammation models and assays for neuropsychiatrically relevant behavioral domains in order to provide a toolbox to study neuropsychiatric comorbidities in skin inflammation on a pre-clinical level, with a focus on mouse models.

### 3.1. Mouse Models for Chronic Skin Inflammation

Many different mouse models for AD and psoriasis have been developed in the past years. Table 2 provides an overview of the more widely used current mouse models. These models are either based on the application of substances that induce an immune response in the skin or mutant mice are used, which lack or overexpress genes associated with the disorder. When using mice, one has to keep in mind that they differ from humans in their epidermal barrier microanatomy and microbiome. For example, densely distributed hair follicles are found in mice, but not in the human skin, and mice express different subtypes of inflammatory and dendritic cells [43,44,45]. An instructive mouse model should, therefore, focus on mimicking the main symptoms and immunologic features observed in AD and psoriasis patients [46]. As proposed by Gilhar et al. [1], AD mouse models should comprise (a) an AD-like epidermal barrier defect with reduced filaggrin expression along with hyperproliferation and hyperplasia; (b) increased epidermal expression of AD-associated chemokines, such as thymic stromal lymphopoietin (TSLP), periostin and/or thymus, and activation-regulated chemokine (TARC; CCL17); (c) a characteristic dermal immune cell infiltrate with the overexpression of key cytokines, such as IL-4, IL-13, IL-31, and IL-33; (d) distinctive “neurodermatitis” features (sensory skin hyperinnervation, defective beta-adrenergic signaling, neurogenic skin inflammation, and triggering or aggravation of AD-like skin lesions by perceived stress); and (e) response of experimentally induced skin lesions to standard AD therapy [1]. In this line, the application of allergens or irritants induces contact dermatitis and such studies suffer from a lack of standardization. Transgenic mice manipulating the key features of AD pathophysiology and immune response avoid these challenges and might be beneficial [1]. 

Mutant mice are also commonly used in psoriasis research. Notably, psoriasis is only naturally occurring in rhesus monkeys [65], cynomolgus monkeys [66], and humans. The first genetic models appeared through a number of spontaneous mutations in mice (see Table 2; [67,68]). As for AD, some models rely on the application of substances inducing acute skin inflammation. Here, the imiquimod (IMQ) mouse model is most commonly used. This model was applied to highlight the correlation of stress and psoriasis exacerbation with increased levels of the neurotransmitter SP, IL-1β, and IL-23p40 [69]. Furthermore, the IMQ model provided important insights into pathomechanisms, e.g., by demonstrating a proinflammatory induction of TNF-α by the fibroblast growth factor (FGF)-7 pathway [70], and helped to discovered the inhibition of Toll-like receptor (TLR) as a treatment for reducing erythema and scales [71]. Transgenic mouse models manipulating specific immunological molecular components help to unravel the pathomechanisms of psoriasis further, suggesting that the observed hyperproliferation of keratinocytes is mediated via the altered regulation of transcription factor signal transducers and activators of transcription 3 (STAT3; [46,63,64]), as well as an aberrant T-cell function [46]. For more details regarding the various psoriasis mouse models, see also Bochenska et al. [44].

Moreover, although hardly any mouse model may imitate all human AD and psoriasis features, the available mouse models are relevant to study the skin–brain axis and their comorbid neuropsychiatric problems as well. Assays for acute and chronic stress exposure in mice are well established and there are several available behavioral protocols, which can help to obtain an experimental handle on the features of neuropsychiatric disorders in mice.

### 3.2. Translational Testing of Neuropsychopathologies in Mice

With several mouse models available to probe for certain features of AD and psoriasis, the research is mostly restricted to the skin. However, the same models can be used to investigate the skin–brain axis in more detail. Testing for neuropsychiatric traits in mice can be performed even with relatively low technical requirements, given a trained experimenter. Here, we introduce some common behavioral tests for depression-, anxiety-, and addiction-like behavior in mice (see Table 3 for overview) that may provide, in combination with AD and psoriasis mouse models, insights into the developments of neuropsychiatric comorbidities in these disorders. 

For all tests, strain variability [1] and confounding factors of rearing (e.g., circadian cycle, ambient noise, interaction with the experimenter, etc.) may also influence several behavioral domains. Therefore, it is important to always compare the behavior to appropriate control groups, e.g., wild-type littermates reared under the same conditions or sham-treated control mice.

#### 3.2.1. Basic Characterization of the Neurological Status

Many behavioral tests rely on the proper functions of the sensory and motor systems. Therefore, it is important to check for the basic neurological status of the mice and motor performance before starting to assess complex behavioral phenotypes regarding emotional control and cognition. An initial screening should comprise tests for normal movement, posture, and reflexes, e.g., with the SHIRPA test [72], and rotarod or beam walking paradigms to test more specifically for motor coordination [74,75,92]. Activity as well as more complex behaviors, e.g., grooming, eating, drinking, and resting time, can be evaluated in a number of automated home cage activity systems (e.g., PhenoTyper, Noldus, the Netherlands; for a comparison of different systems see [73]), which allow assessing circadian rhythms as well. The open field (OF), a small square arena, is also commonly used to evaluate locomotion and spontaneous exploratory behavior in mice. Many commercial and open-source programs are available to automatically analyze video recordings from such a test session and gather data regarding running pattern and locomotor activity [76]. 

#### 3.2.2. Anxiety Testing

The open field can be used further to measure anxiety-like behavior. Tests for anxiety utilize internal conflicts between exploring an environment with the prospect of finding food or social partners and the danger of being exposed to potential predators. Consequently, more anxious animals would spend less time in exposed areas, such as the center of an open field [76,79], arms without protective walls in a cross-shaped elevated plus maze (EPM) [77,79], or the brightly lit compartment of a light–dark box [78]. Neophobia, the fear of novel objects, can be assessed by introducing marbles to a standard cage and counting how many of them will be actively covered by bedding in a given time period (marble burying test, MBT). While the excessive burying of marbles is rated as anxiety-like behavior, it is also used to assess repetitive and compulsive behavior [79]. 

#### 3.2.3. Depression

Major depression in humans is characterized by a sustained “depressive state” with a diminished interest in pleasure and activities, fatigue, feelings of worthlessness, and indecisiveness, accompanied by sleep and concentration disturbances [93]. Anhedonia, or the loss of interest in pleasures, is one of the most often classified depression-like behavioral states in mice. It can be verified using the sucrose preference test, where depressive-like behavior would be indicated by a reduced choice of drinking water containing sweetener [80]. Another core symptom, social withdrawal, can be tested with social interaction paradigms, e.g., the three-compartment test. Here, mice can choose to explore a compartment containing a conspecific partner mouse vs. an empty compartment or an unanimated object, or they can stay in a central compartment without any interactions at all. By introducing new interaction partners, the test can be modified for social memory features [81]. Impairments of daily life activities can simply be tested by assessing nest building, i.e., scoring the complexity of the nest built from a paper tissue within 24–48 h. Disturbed nest building is observed in several neuropsychiatric disorders models, including obsessive compulsive disorders (OCDs) [82,94]. Widely used, but critically discussed, is the use of the Porsolt forced swim test in depression models. Here, the mouse is placed into a tub filled with lukewarm water and a lack of swimming activity is often interpreted as despair-like behavior. While such immobility is quickly reduced by treatment antidepressants, the validity of the test is questionable and may rather reflect learning coping strategies instead of behavioral despair (see [95,96] for an in-depth discussion).

#### 3.2.4. Addiction

Alcohol abuse is an addiction often comorbid with AD and psoriasis and can be triggered by stress exposure. To test whether this also applies in the respective mouse models, a test for voluntary alcohol intake can be conducted. The majority of the studies use two-bottle choice (2BC) tests of ethanol vs. water, but sometimes even different concentrations of ethanol are offered in parallel (e.g., four-bottle choice (4BC) test) [97]. Commercially available drinkometer systems allow for a high-resolution monitoring of alcohol drinking patterns [83]. 

#### 3.2.5. Learning and Memory

Reduced cognitive abilities are a common feature of many neuropsychiatric disorders, such as depression, and an altered memory for aversive events is one of the core symptoms of post-traumatic stress disorder, a special anxiety disorder. Signature tests for memory capacity in mice include spatial or novel object recognition tests (ORTs), which are based on the rodents’ innate preference for novelty. In the novel object version, mice have to discriminate between familiar and newly introduced objects inside an OF arena, while in the spatial version, one familiar object is moved to a new location [84,85]. Spatial memory capacity can also be investigated using the water maze or a Barnes maze, where animals learn to navigate to specific locations to escape from water or a brightly lit environment [98]. Alternatively, mice can also build a spatial memory to navigate to a food reward and use a radial maze usually containing eight different arms. Once established, new locations can be introduced, which would require reversal learning and even more complex learning rules than what can be investigated in such mazes [87]. Remembering recently visited arms in a radial maze requires further functioning of a working memory to execute a cognitive task correctly. Working memory depicts a prerequisite for decision making and is disturbed by several neuropathologies, including Alzheimer’s disease [99]. It can be tested in simpler T- or Y-shaped setups and can involve delays for remembering the latest arms visited to enhance cognitive demands (delayed matching to sample, DMTS) [88,100]. Even more complex memory tasks comprise serial-choice tasks, e.g., the five-choice serial reaction time task (5CSRTT), where rewards are associated with the sequences of specific stimuli [89]. 

To study the molecular mechanisms of memory formation, fear conditioning is one of the most commonly used paradigms, since it induces a precise and stable aversive memory by associating a stimulus or environment with a threatening foot shock [101]. In rodents, defensive behavior upon re-exposure to the environment (the context) or the stimulus can be easily recognized by video tracking systems and quantified in rodents by measuring freezing time, i.e., the absence of movement, except for respiration [90]. Notably, increased fear memory and a generalization of the memory to other stimuli (e.g., auditory stimuli with different frequencies or neutral environments) have been observed in certain anxiety disorders, such as phobia or post-traumatic stress disorder [91]. 

The behavioral tests briefly introduced here focus on different brain circuits. The hippocampus is especially involved in spatial memory, contextual fear memory, and with its ventral subportion also in anxiety. More complex tasks and reversal learning rather involve the prefrontal cortex. Learned fear and anxiety, but also addiction, is mediated by circuits involving the amygdala and the nucleus accumbens/ventral tegmental area. Importantly, shifts in the functional connectivity between these brain areas are commonly observed in patients with anxiety disorders, depression, and addiction, providing valuable translational power for the behavioral tests described here. Specialized setups for the given types of tests are nowadays commercially purchasable and offer state-of-the-art video tracking as well as an automated assessment of a variety of data. First pilot projects suggest the artificial intelligence (AI)-based big data analysis of mice performing tests and social interactions in more naturalistic settings (e.g., a “mouse city”). This might offer revolutionary, and most of all unbiased, insights into the mouse behavior of wild-type and transgenic lines; however, it will presumably be more difficult to translate to symptom clusters in neuropsychiatric patients.

By combining some of the mentioned behavioral tests with video tracking and in-depth analysis, a convenient test battery to behaviorally characterize mouse models for AD and psoriasis can be compiled. This is especially interesting in respect of the comorbidities and will produce a meaningful outline of the bodily and mental wellbeing of the animals. 

## 4. Interventional Approaches and Translational Relevance for Probing the Skin–Brain Axis

To break the vicious cycle of aggravating skin symptoms by reduced mental wellbeing with stress triggering skin symptoms, new therapeutic strategies need to be developed to also treat the neuropsychiatric comorbidities in AD and psoriasis. Characterizing features of neuropsychiatric comorbidities in mouse models of AD and psoriasis can be one of the primary steps to better understand the skin–brain axis in these diseases. However, in order to mechanistically link underlying molecular events and circuits, strategies for neuronal interventions are required and their translational perspectives needs to be explored.

### 4.1. Interventional Approaches

The past decades have equipped researchers with an ample treasure trove of tools for genetic manipulations. The murine model strongly benefited from the invention of the Cre-loxP system, which allows for cell-specific modifications targeting skin, immune, and brain cells [102]. In addition, chemogenetics and optogenetics provide the opportunity of remote-controling neural activity (facilitation or silencing, respectively) at a higher spatial resolution than ever before in cells of interest [103]. In the case of chemogenetics, DREADD (Designer Receptors Exclusively Activated by Designer Drugs) constructs are transferred to cells by viral vectors. The then-expressed engineered proteins interact with previously biological inactive small molecules, allowing for a time-restricted modification of cellular activity [103]. Optogenetics offers an even more precise temporal resolution by inhibiting or activating the neural activity in specific brain regions in response to light via expressing light-sensitive ion channels, pumps, or enzymes [104]. To monitor cellular activity during behavioral tasks, the levels of promoters of immediate early genes (IEGs), such as Fos or Arc, can be evaluated [105,106], or transient elevations of intracellular Ca^2+^ concentrations can be measured [107]. New tools, such as opto- and chemogenetic vectors under the control of cFos promotor constructs, even allow us to manipulate only such activated cells and deliver valuable insights into circuits relevant for memory formation [101]. If the relevant target cells are known, retrograde tracing to label presynaptic neurons can be a powerful technique [108,109].

With these tools for spatially and temporally precise manipulations in vivo at hand, and a whole arsenal of transgenic mouse lines [110], the remaining hurdle is to identify the actual target cells and molecular substrates. To gain further insights into a putative skin–brain axis, it could be of interest to target microglia and investigate immune-specific cell activation in the brain versus the periphery. Problematically, microglia cannot be easily differentiated from border-associated macrophages (BAMs) and peripheral myeloid cells. (For more details of various means for differentiation see, e.g., [111,112,113]). The most promising mouse lines are based on microglia signature genes and are either engineered as knock-in strains with fluorescent reporter proteins being largely restricted to microglia, or as inducible Cre lines, with Hexb^CreERT2^ mice having the highest specificity for microglia [112]. Other inducible Cre lines are available to achieve microglia depletion. For example, in Cx3cr1^CreER^:R26^iDTR^ mice, microglia expresses the diphtheria toxin receptor, with the result that, after the administration of diphtheria toxin, microglia is depleted within one day and the effect lasts for up to seven days [114]. 

Another target in the investigation of the skin–brain axis could be peripheral nerve signals to the central nervous system. In order to target only the peripheral nerve, but not spinal microglia, manipulations could be implemented at the level of the dorsal root ganglion (DRG), where the cell body of the pseudo-unipolar sensory neuron is located [115]. Manipulations of the DRG have largely been reported in the context of traumatic injury and include, for example, the chemogenetic activation (using AAV5-hSyn-hM3Dq-mCherry) of DRG sensory neurons in vitro and in vivo [116], or the direct injection of anti-inflammatory mediators into the DRG to investigate its importance in pain signaling [117]. 

To gain more insights in the local effects of immunomodulators, a local and cell-type-specific overexpression or knock down of interleukins in skin and immune cells and in various brain areas can be conducted. One example could be the use of specific Cre-driver lines to entangle the effects of glia-specific overexpression versus T-cell-specific overexpression of a respective interleukin. Another approach could be a local overexpression of relevant interleukins, such as IL-4, in the hippocampus, and PFC in combination with spatial and working memories or a knock down of this factor in animals with chronic cutaneous inflammation. As there are many open ends to probe for, further investigations can focus on the more central or peripheral parts of the skin–brain axis and help to understand their differential role in pain signaling and inflammatory events.

### 4.2. Translational Outlook

As discussed above, the mouse model depicts an interesting and genetically tractable study case with a large research community providing steady innovations [104,118]. Despite some differences in the skin composition, immune system, and development of neuronal circuits, in comparison to humans, mice are suitable for translational studies. They provide a good compromise between simplicity in terms of cell numbers and conservation of genetic and cellular properties [119,120]. Mice have been widely applied to study the connectivity within the brain, but also peripheral systems in both homeostatic and diseased states. They have been especially worthwhile to expand our knowledge on the immune system and shed light on many common principles, including T-cell receptors, histocompatibility complex genes, and regulation of antibody synthesis, to name but a few [121,122]. Particularly, interleukin markers are commercially available that allow for studying their function in the skin–brain axis in the mouse model. This is important because measures of interleukin serum levels in human patients have, to date, proven to be inconclusive. Novel insights into the skin–brain axis of mice have the potential to spark further research in the human model and encourage rethinking therapeutic approaches of inflammatory diseases. Future pharmaceutical and therapeutic approaches should consider all parts of the brain–skin axis involved in the inflammatory events, instead of focusing on only one part of the system and neglecting the rest. Only a holistic approach is likely to break the vicious cycle of peripheral and central inflammatory processes in the long run. 

## 5. Conclusions and Future Perspectives

Many tools and models are available to study the (dys-) function of the skin, the immune system, and the brain in chronic cutaneous inflammatory disease, such as AD and psoriasis. However, complex interactions between systems, such as in the skin–brain axis, are less well studied, although observations in patients firmly establish that such a link exists. Here, we proposed a procedure based on a combination of manipulation, recording, behavioral tasks, and cellular analysis in mouse models of AD and psoriasis (for review of such a course of action, see Nakajima and Schmitt [123]). Although it is hardly possible for animal models to exactly mimic the human disease under investigation in every aspect, they provide valuable information on the pathomechanism of the disease. Thus, expanding the characterization of AD and psoriasis mouse models to behavioral entities relevant for neuropsychiatric comorbidities can hold valuable results with translational value for the human condition. Such an approach opens the possibility to identify molecular targets strongly mediating neuropsychiatric effects in these disorders and to engineer novel therapeutic strategies. In addition, the ample repertoire of genetic tools allows for time- and celltype-specific manipulations, leading to a better understanding of molecular events on the cell and circuit level. Moreover, protein and gene expression analyses in combination with computational modeling will help to analyze shifts in complex systems upon such manipulations (see also Figure 3). Eventually, multidisciplined approaches will help to update mouse models and increase their translational relevance to model the skin–brain axis and pave the way for further studies of peripheral and central inflammatory events and beyond. Importantly, investigations of the bidirectional impact of peripheral and central systems in cutaneous inflammatory diseases will help to identify resilience factors that can later be translated into drugs against inflammatory events. One putative resilience factor is neuropeptide Y (NPY), a key resilience factor of the hippocampus. Towards this end, existing animal models could be used in the above suggested combinatorial approach, for example, by activating NPY receptors in distinct brain areas in mouse models with chronic inflammation. This approach could elucidate whether the severity of skin inflammation is indeed modulated by NPY [124,125]. Similarly, tissue samples should be screened for other potential resilience factors to boost the development of novel drugs, which do not only exclusively focus on the improvement of skin inflammation, but rather do so by strengthening central brain resilience. Resilience will in turn improve stress responses, and thereby reinforce the stress axis and its underlying factors.

## Figures and Tables

**Figure 1 ijms-23-07484-f001:**
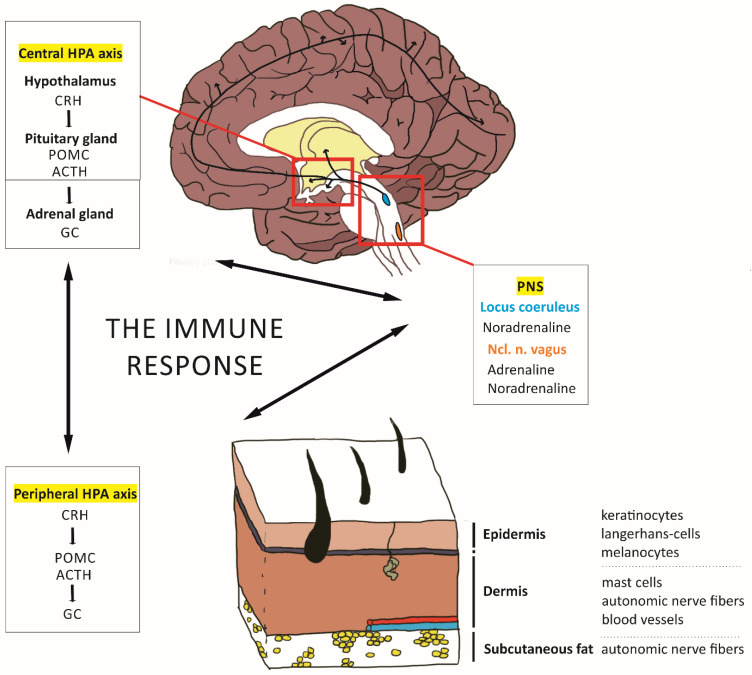
An immune response requires the interaction of the central HPA axis, another peripheral HPA axis, and the PNS. The central HPA axis reacts to stressors with the release of CRH from the paraventricular nucleus of the hypothalamus, which in turn leads to the secretion of POMC and ACTH from the pituitary gland. ACTH subsequently induces an increase in GC levels in the adrenal gland. Correspondingly, the peripheral HPA axis of the skin enacts a similar cascade. CRH is released from nerve fibers, or cells of the skin and leads to a release of POMC and ACTH, which again results in elevated GC levels. In principle, three different cell types of the skin play a major role for the immune response. These include keratinocytes and melanocytes, which reside in the epidermis, and mast cells, which are located in the dermis. Their differentiation and proliferation status are tightly regulated by CRH; however, disturbances in the HPA axis and, thus, CRH levels can easily lead to crucial changes in cell fate. Both the central and peripheral HPA axis are input and respond to the PNS. Most notably, the locus coeruleus is a main source of noradrenaline and exerts its influence on the limbic system as well as the mPFC and the ACC. In addition, the Ncl. n. vagus of the brainstem controls the release of adrenaline and noradrenaline in response to stressors.

**Figure 2 ijms-23-07484-f002:**
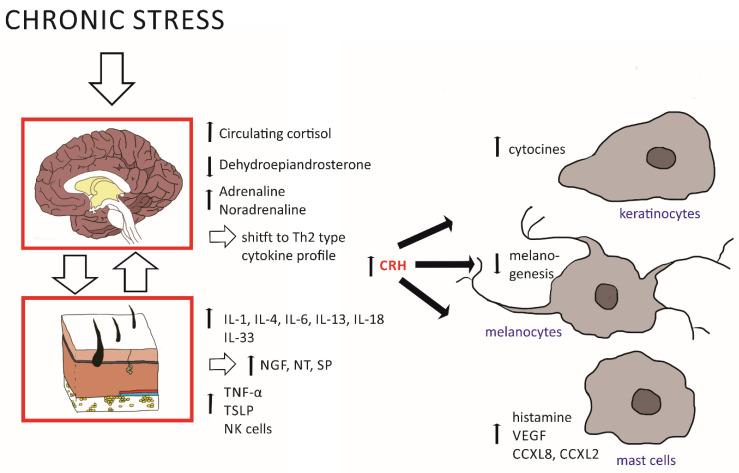
Chronic stress acts on the central as well as peripheral systems to induce adequate immune responses. An increase in circulating cortisol and levels of adrenaline and noradrenaline together with a decrease in dehydroepiandrosterone leads to a Th2-type shifted cytokine profile. In both the central and peripheral HPA axes, CRH and its downstream cascade are elevated. This leads to a release of TNF-α, TSLP, and of pro-inflammatory cytokines, such as IL-1, IL-4, IL-6, IL-13, IL-18, and IL-33. Especially, IL-33 is of major impact as it triggers an increase in NGF, NT, and SP levels. NT, in turn, acts on mast cells to induce the release of histamine, while SP activates both keratinocytes and mast cells and causes the release of VEGF from the latter. The activation of the respective cells of the skin leads to the release of even more CRH, which exacerbates inflammatory processes by causing malfunctions of the skin barrier, decreasing melanogenesis, and thus increasing permeability.

**Figure 3 ijms-23-07484-f003:**
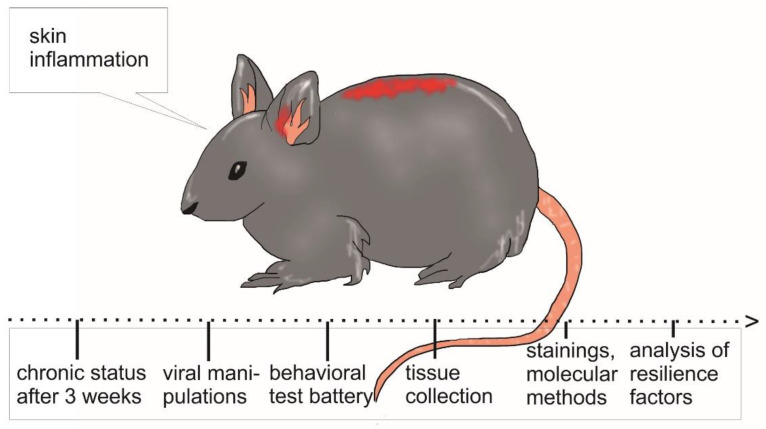
Proposed mouse model for cutaneous inflammation. After inducting a chronic skin inflammation viral intervention, e.g., chemogenetic manipulations can be activated and the animals are subjected to a behavioral test battery, including signature tests for locomotion, anxiety, depression, and learning and memory. Subsequently, tissue samples are collected from the animals and further investigated using immunohistochemistry and molecular methods. The processed samples are then analyzed for changes in inflammatory factors and stress-associated hormones and brain peptides.

**Table 2 ijms-23-07484-t002:** Mouse models to probe the skin–brain axis. Mouse models for atopic dermatitis (AD) and psoriasis.

(A) AD	Description	Advantages/Caveats	Reference
Oxazolone (OXA) application	Destroyed integrity of skin barrier Th2 immune response	(+) rapid, low cost (−) model for allergic contact dermatitis	[1,47,48]
Chicken-egg albumin-ovalbumin (OVA) application	Triggers Th2 immune response	(+) chronic AD-like skin lesions (−) variable OVA allergen composition (−) not sufficient in certain mouse strains (e.g., C57BL/6)	[1,49,50,51]
Calcipotriol (MC903) application	Activation of ILC2 type-2 immune response with eosinophilia, skin swelling, inflammation	(+) model for type 2 immune response initiation (+) to study TSLP and neutrophils in scratching behavior	[52,53,54,55,56]
Flg ft/ft or Flg -/- “Flaky tail mice”	Filaggrin deficient mice	(+) spontaneous dermatitis (+) to study skin microbiome	[57,58]
Blmh -/- mice	Bleomycin hydrolase (BLMH) deficiency impairs filaggrin processing	(+) decreased levels of natural moisturizing factors (+) decreased levels of BLMH in AD	[58]
Interleukin overexpression	Overexpression of IL-4, IL-5, IL-13, IL-18, IL-31, TSLP	(+) exploration of specific pathways	[47]
Imiquimod application	Acute skin inflammation	(+) erythema and scales as in human disorder (+) used to study stress–skin symptom correlation	[59]
Ttc7 fsn/Ttc7 fsn	Spontaneous mutation in tertratricopeptide repeat domain 7	(+) progressive papulosquamous as in human disease	[60]
cpdm/cpdm	Spontaneous proliferative dermatitis mutation mouse	(+) red and scaling skin as in human disease	[61]
Scd1 ab/Scd1 ab	Asebia mouse, defective stearoyl-CoA desaturase-1 (Scd1) gene	(+) leads to hypoplastic sebaceous glands	[62]
Interleukin signaling	Overexpression/knock out of IL-6, IL-20, STAT3 pathway	(+) hyperproliferation of keratinocytes and altered differentiation via STAT3 pathway	[46,63,64]
Transgenic mice for aberrant T-cell function	Via TGF ****** regulating T cell development	(+) altered keratinocyte regulation	[46]

****** TGF: transforming growth factor.

**Table 3 ijms-23-07484-t003:** Behavioral assays for neuropsychiatric features in mice.

Assay	Read-Outs	Associated Psychiatric Feature	Reference
SHIRPA test	Movement, posture, reflexes	Basic neurological characterization	[72]
PhenoTyper	Activity	Circadian rhythm, basal activity	[73]
Rotarod	Motor learning	Neurological motor and coordination deficits	[74]
Beam walking	Motor coordination	Neurological motor and coordination deficits	[75]
Open field	Time and distance covered	Locomotory activity, anxiety	[76]
Elevated plus maze	Time, distance, and entries in open and closed arms	Locomotory activity, anxiety	[77]
Light–dark box	Transitions between compartments, time spent in compartments	Anxiety	[78]
Marble burying test	Numbers of marbles covered by bedding	Anxiety, compulsive behavior, repetitive behavior	[79]
Sucrose preference test	Consumption of plain water vs. water with sweetener	Anhedonia, depression	[80]
Social interaction test	Time contacting a social interaction partner restricted in a tube	Social preference, social anhedonia, social memory	[81]
Nest building	Complexity scores of nests built from tissues	Reduced wellbeing, depression, compulsive and repetitive behavior	[82]
Two-bottle choice test	Consumption of plain water vs. ethanol	Addiction	[83]
Object recognition	Time spent with familiar vs. novel objects or object locations	Recognition and spatial memory	[84,85]
Water maze/Barnes maze	Latency to reach an escape platform/hole, time spent at the escape platform/hole	Spatial memory	[86]
Radial arm maze	Latency to reach a reward arm	Spatial memory, working memory	[87]
Delayed matching to sample test	According to a learning rule correctly entering a specific arm on a T-maze after a delay	Working memory	[88]
5CSRTT	Correct choices for retaining a reward associated with sequences of stimuli	Working memory, attention, impulsivity	[89]
Fear conditioning	Freezing to a conditioned stimulus or context	Aversive memory	[90,91]

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
