# Peer review of "Probing the Skin–Brain Axis: New Vistas Using Mouse Models"

_ijms, 2022, doi:10.3390/ijms23137484_

Round 1

Reviewer 1 Report

The authors reviewed a new study on the brain-skin correlation. The possibility of verifying this through an animal model was presented. With a few modifications, it is worthy of publication in IJMS.

major point

1. The title suggests that the animal model is the main content of this thesis. It would be nice if you could draw a proposed animal model that is easy to recognize with a figure.

2. Rather than presenting a new animal model that verifies the skin-brain-axis, the skin inflammation model and the brain research model are described at length. Rather, it is necessary to present a new animal model that may interest readers.

3. Suggesting a strategy for developing new drugs that can improve skin diseases by controlling psychology will be good information for readers.

Author Response

We thank the reviewers for the encouraging and positive comments on our manuscript.

We addressed the reviewer’s critical points and suggestions in a point-by-point manner below.

Reviewer 1:

The authors reviewed a new study on the brain-skin correlation. The possibility of verifying this through an animal model was presented. With a few modifications, it is worthy of publication in IJMS.

major point

  1. The title suggests that the animal model is the main content of this thesis. It would be nice if you could draw a proposed animal model that is easy to recognize with a figure.

Indeed, our review focuses on animal models – in combination with interventional approaches to study underlying molecular factors and pathways in more detail and to investigate neglected interactions between central and peripheral pathways. In order to make the course of action of the proposed combinatorial approach more descriptive, we followed your suggestion to include a respective figure (Figure 3, line 559-566). We hope you find this visualization helpful for readers and follow the timeline/course of action of the approach with our proposed mouse model for cutaneous diseases.

  1. Rather than presenting a new animal model that verifies the skin-brain-axis, the skin inflammation model and the brain research model are described at length. Rather, it is necessary to present a new animal model that may interest readers.

 Thank you for the critical comment regarding the focus of this review. Indeed, our aim/focus was to suggest a combinatorial approach (see 4. ‘Conclusions and future perspectives’, line 593-606) to study inflammatory skin diseases. As described in the review, mouse models exist based on molecular factor associated with the skin diseases, but they are rarely investigated for their behavioral phenotypes and changes in plasticity-relevant molecular factors in brain areas involved in cognition and emotion. Instead of developing completely new models, it is rather timely to combine existing expertise across research fields. In this review, we therefore aim for alerting the reader to the fact that a combination of animal models, genetic tools and manipulation techniques, protein and gene expression analyses is crucial to allow for a holistic study of the inflammatory factors involved and in turn their impact on behavior. Moreover, the combination with inducible interventions of pathways and molecular factors, e.g., by using viral vectors, could reveal causal coherences with, for example, microglia, peripheral or central cytokines and resilience factors against inflammatory processes. That would not exclude the evolution of mouse models in future by developing our cross-sectional approach further, e.g., by including stress resilience factors (see also response to comment #3 and discussion line 609-620).

  1. Suggesting a strategy for developing new drugs that can improve skin diseases by controlling psychology will be good information for readers.

 This point is indeed interesting and important and we follow up on it now in part 4. ‘Conclusions and future perspectives’ (line 609-620). Here, we suggest that the major output of a combinatorial study, as we propose it, is the identification of resilience factors which are known to strengthen central and peripheral responses to chronic stress. A possible candidate for is neuropeptide Y (NPY), which is already known as resilience factor of the hippocampus. We propose further studies to investigate whether NPY also modulates the severity of skin inflammation and to identify additional resilience factors which can be useful for drug research.

Reviewer 2 Report

The topic of this review is extremely interesting and it can be seen that it's a result of a thorough research. However, the description of the methods for examining the neurological and psychological status of mice is disproportionately long. A shortening of this section and a more detailed description of the signaling molecules/pathways involved in skin-brain communication would make the article much better.

Minor misspellings in the text:

line 241: What is the abbreviation for "AS"? Didn't you want to write "AD"?

line 243, 519 and 526: Psoriasis is written with "P", in the rest of the text is written with "p"

line 268: What does the word "bus" mean in this context?

line 337: "the" is written instead of "they"

Author Response

We thank the reviewers for the encouraging and positive comments on our manuscript.

We addressed the reviewer’s critical points and suggestions in a point-by-point manner below.

Reviewer 2:

The topic of this review is extremely interesting and it can be seen that it's a result of a thorough research. However, the description of the methods for examining the neurological and psychological status of mice is disproportionately long. A shortening of this section and a more detailed description of the signaling molecules/pathways involved in skin-brain communication would make the article much better.

We shortened now part 2.2. ‘Translational testing of neuropsychopathologies in mice’ considerably and added Table 3 (line 303 and following) in order to provide an overview of the various behavioral tests in a more concise manner (line 295).

In addition, we followed your suggestion to provide more detail regarding the molecular pathways in skin-brain interactions in part 1.2 ‘Effects of peripheral inflammation on brain plasticity and stress-related behavioral domains’ inflammation (line 156-178). Here, we describe known mechanisms of action of cytokines in the brain, as well as the connection of central brain inflammatory processes with peripheral processes. We also describe potential pathways of signature type 2 inflammatory cytokines in the introduction (line 28-34). We hope that you will find the additional information sufficient to round up the picture of peripheral and central.

We corrected the spelling mistakes according to your suggestions:

Line 241 (now line 259): ‘AS’ was changed to ‘AD’

Line 243, 519 and 526 (now line 261 and the following): the P in psoriasis was changed to small letters

Line 268 (now line 286): the word ‘bus’ was changed to ‘be’

Line 337 (now line 374): the word ‘the’ was changed to ‘they’

Round 2

Reviewer 2 Report

The authors complied with all previously formulated suggestions. I hope they also feel that the article they wrote has become much more professional.